# Effects of *Escherichia coli* LPS Structure on Antibacterial and Anti-Endotoxin Activities of Host Defense Peptides

**DOI:** 10.3390/ph16101485

**Published:** 2023-10-18

**Authors:** Ali Javed, Melanie D. Balhuizen, Arianne Pannekoek, Floris J. Bikker, Dani A. C. Heesterbeek, Henk P. Haagsman, Femke Broere, Markus Weingarth, Edwin J. A. Veldhuizen

**Affiliations:** 1Section of Immunology, Division of Infectious Diseases and Immunology, Department of Biomolecular Health Sciences, Faculty of Veterinary Medicine, Utrecht University, 3584 CL Utrecht, The Netherlands; a.javed@uu.nl (A.J.); a.a.pannekoek@uu.nl (A.P.); f.broere@uu.nl (F.B.); 2NMR Spectroscopy, Bijvoet Centre for Biomolecular Research, Department of Chemistry, Faculty of Science, Utrecht University, 3584 CS Utrecht, The Netherlands; m.w.weingarth@uu.nl; 3Section of Molecular Host Defense, Division of Infectious Diseases and Immunology, Department of Biomolecular Health Sciences, Faculty of Veterinary Medicine, Utrecht University, 3584 CL Utrecht, The Netherlands; m.d.balhuizen@uu.nl (M.D.B.); h.p.haagsman@uu.nl (H.P.H.); 4Department of Oral Biochemistry, Academic Centre for Dentistry Amsterdam, University of Amsterdam and VU University Amsterdam, 1081 LA Amsterdam, The Netherlands; f.bikker@acta.nl; 5Department of Medical Microbiology, University Medical Centre Utrecht, 3584 CX Utrecht, The Netherlands; d.a.c.heesterbeek-2@umcutrecht.nl

**Keywords:** host defense peptides, LPS, Lipid A, O-antigen, antimicrobial resistance

## Abstract

The binding of Host Defense Peptides (HDPs) to the endotoxin of Gram-negative bacteria has important unsolved aspects. For most HDPs, it is unclear if binding is part of the antibacterial mechanism or whether LPS actually provides a protective layer against HDP killing. In addition, HDP binding to LPS can block the subsequent TLR4-mediated activation of the immune system. This dual activity is important, considering that HDPs are thought of as an alternative to conventional antibiotics, which do not provide this dual activity. In this study, we systematically determine, for the first time, the influence of the O-antigen and Lipid A composition on both the antibacterial and anti-endotoxin activity of four HDPs (CATH-2, PR-39, PMAP-23, and PMAP36). The presence of the O-antigen did not affect the antibacterial activity of any of the tested HDPs. Similarly, modification of the lipid A phosphate (MCR-1 phenotype) also did not affect the activity of the HDPs. Furthermore, assessment of inner and outer membrane damage revealed that CATH-2 and PMAP-36 are profoundly membrane-active and disrupt the inner and outer membrane of *Escherichia coli* simultaneously, suggesting that crossing the outer membrane is the rate-limiting step in the bactericidal activity of these HDPs but is independent of the presence of an O-antigen. In contrast to killing, larger differences were observed for the anti-endotoxin properties of HDPs. CATH-2 and PMAP-36 were much stronger at suppressing LPS-induced activation of macrophages compared to PR-39 and PMAP-23. In addition, the presence of only one phosphate group in the lipid A moiety reduced the immunomodulating activity of these HDPs. Overall, the data strongly suggest that LPS composition has little effect on bacterial killing but that Lipid A modification can affect the immunomodulatory role of HDPs. This dual activity should be considered when HDPs are considered for application purposes in the treatment of infectious diseases.

## 1. Introduction

Host Defense Peptides (HDPs) are antimicrobial molecules that are part of the innate immune system. They are short, cationic, and amphipathic peptides [1], and they often exert their antimicrobial activity by targeting bacterial membranes. Different models have been proposed to describe this membrane interaction, all eventually resulting in bacterial lysis [2]. The first step in several models is the interaction between the cationic parts of the HDPs and the anionic lipopolysaccharide (LPS) molecules in the outer membrane of Gram-negative bacteria [3]. This electrostatic interaction could facilitate the subsequent hydrophobic interaction with acyl chains and thereby facilitate insertion into or translocation through the bacterial membrane. In addition to bacterial killing, HDPs are also involved in neutralizing the toxic effects of LPS. Interaction of LPS with Toll-Like Receptor 4 (TLR4) on immune cells, such as macrophages, leads to strong pro-inflammatory activation of these cells, which can lead to severe tissue damage and, in extreme cases, sepsis. HDPs are thought to play a role in the inhibition of this LPS-induced pro-inflammatory response [4,5]. The concept that certain HDPs can kill bacteria and subsequently neutralize an unwanted LPS-mediated proinflammatory immune reaction towards released LPS is known as ‘silent killing’ and was described in detail for chicken cathelicidin 2 (CATH-2) [6]. It provides HDPs such as CATH-2 with a major advantage when they are considered alternatives to conventional antibiotics.

LPS is the main molecule in the outer leaflet of the outer membrane of Gram-negative bacteria. It consists of three domains: the conserved lipid A, the core domain, and the O-antigen that protrudes into the extracellular environment [7]. The lipid A portion consists of acyl chains attached to a phosphorylated N-acetylglucosamine (NAG) dimer. Attached to lipid A is the core moiety, which is an oligosaccharide (mostly six to twelve sugar moieties) that includes common sugars and sugars that are unique to bacteria. The outermost part of the LPS is the O-antigen, a polysaccharide which consists of repeating sugar subunits, comprised of one to five different sugars. Importantly, bacteria can modify their LPS to adjust to their environment. This involves Lipid A as well as non-Lipid A modifications, interestingly also providing extra protection by immune detection evasion or by conferring resistance to certain antibiotics [7]. Although the main lipid A structure is conserved, some unusual Lipid A structures with modified phosphate groups or varying number/length of acyl chains can be found in some species [8]. For instance, as an important resistance mechanism towards antibacterial compounds such as colistin and polymyxin B, some bacteria, mainly *Escherichia (E.) coli*, alter the phosphate groups of Lipid A with the addition of a phosphoethanolamine (pEtN) residue by phospho-form transferases like EptA or MCR-1 [9]. The length of the O-antigen is an important modification as well; it can vary up to forty repeats of sugar subunits, while in other strains this O-antigen is completely absent. LPS without the O-antigen is often referred to as lipo-oligosaccharide (LOS) [10]. LPS containing an O-antigen is called smooth, while LPS without an O-antigen is called rough. Changes in position and numbers of saccharides in the core region of *E. coli* LPS have been shown to affect bacterial membrane permeability and survival [11]. All variations present in the LPS structure could, in theory, affect HDP antibacterial and antiendotoxin effectivity, but especially the latter has not been intensively studied [12].

It is important to specify the exact strains used in this study since LPS structures between *E. coli* strains can already differ substantially. *E. coli* O111:B4 LPS consists of six acyl chains, two Kdo moieties with two phosphates in Lipid A, and an O-antigen of four to forty repeats of a five-sugar moiety, identical to that of *Salmonella enterica* O35 [13,14,15,16]. *E. coli* K-12 LPS has an identical lipid A moiety as *E. coli* O111:B4 but lacks the O-antigen and differs slightly in the core sugar [17]. *E. coli* K-12 can possess a third Kdo moiety or a rhamnose attached to the two Kdo moieties. Furthermore, where the outer core sugar of *E. coli* K-12 contains three glucose moieties, one galactose and a heptose, the outer core sugar of *E. coli* O111:B4 contains a glucosamine instead of a heptose and a slightly different linkage [18,19]. MCR-1^+^
*E. coli* was included to test its cross-resistance to the tested HDPs. Monophosphoryl Lipid A (MPLA), which lacks one of the two typically present phosphate groups, and diphosphoryl Lipid A (DPLA) from *E. coli* F583 were included as corresponding Lipid A-modified LPS structural candidates.

Since LPS is the initial contact site for HDPs, its structure and charge could be critical for the susceptibility of Gram-negative bacteria to these antimicrobial peptides [20]. However, the mode of action of these peptides would define if the stronger affinity for LPS is aiding or hindering the membrane disruption. Though the initial interaction of HDPs with LPS is assumed to be advantageous for HDP function, it is also suggested that LPS can actually inhibit peptides from exhibiting their antimicrobial function. It could perform so by strongly binding to HDPs and thereby preventing access to the bacterial inner membrane. Furthermore, LPS could be excreted in outer membrane vesicles (OMVs) from the membrane and thereby even function as a decoy target for HDPs. It was shown for *E. coli* that the addition of isolated OMVs protects the bacteria from killing by CATH-2, PMAP-36, and LL-37 [21], which are HDPs from chicken, porcine, and human origin, respectively. Likewise, OMVs of *Helicobacter pylori* protected the bacterium against LL-37 [22]. It is interesting to study the effect of Lipid A and non-Lipid A modifications of *E. coli* LPS on the bactericidal as well as anti-inflammatory properties of HDPs. This could contribute to the research for the discovery and development of compounds with enhanced activities against *E. coli* infections.

In this study, the binding of four HDPs to *E. coli* LPS was evaluted in detail. The presence of characteristic LPS modifications was correlated to both aspects of the silent killing concept: the antibacterial and LPS neutralization activities of HDPs. The four HDPs studied here were CATH-2 and PMAP-36, both membrane-active peptides [23,24], PR-39, a non-membrane disrupting HDP [25,26,27], and PMAP-23, a membrane-active peptide, but one that was shown not to interact with *Salmonella* Minnesota LPS [28,29,30]. The results obtained indicate that the O-antigen has minimal effect on any aspect of HDP activity, but that modification of the lipid A phosphate group can specifically affect the immunomodulatory effect of certain HDPs.

## 2. Results

### 2.1. Influence of O-Antigen in Resistance to Host Defense Peptides

To determine the effect of the presence of an O-antigen on HDP antibacterial activity, two *E. coli* strains were used, *E. coli* O111:B4 and *E. coli* K-12, with LPS structures that only slightly differ in the core structure but differ mainly in the presence or absence of the O-antigen. The susceptibility of the two strains to a concentration range of HDPs was investigated by MBC analysis using track dilution assays. Only minor differences were observed; *E. coli* O111:B4 was slightly more susceptible to HDP killing than *E. coli* K-12 for CATH-2 and PMAP-36 (Figure 1A,B). This suggests that the presence of O-antigen on *E. coli* O111:B4 has no major impact on HDP antibacterial activity. For a better comparison of antibacterial HDPs, their activities are also shown on a weight-based scale in Appendix A.

### 2.2. Influence of Lipid A Modification in Resistance to Host Defense Peptides

Next, the effect of the Lipid A modification on HDP activity was determined. Colistin-resistant, commercially available *E. coli* NCTC 13864 was compared with clinical isolate *E. coli* 078. The colistin-resistant *E. coli* strain has a modified Lipid A moiety containing a phosphoethanolamine attached to its 4′ phosphate (Figure 2A). MBC values confirmed that the MCR-1-carrying *E. coli* strain was less susceptible to Colistin (Figure 2B). The results showed that there was no observable MCR-1-mediated resistance to CATH-2 and PMAP-23 (Figure 2C,F), with only very small (2-fold) differences in MBCs for PMAP-36 and PR-39 (Figure 2D,E). This suggests that there is no major impact of Lipid A phosphate group modification on the antibacterial activity of the different HDPs. For a better comparison of antibacterial HDPs, their activities are also shown on a weight-based scale in Appendix A.

### 2.3. Neutralization of LPS or Lipid A Induced Macrophage Activation by Host Defense Peptides

LPS, or Lipid A, can activate macrophages by TLR4 activation and trigger downstream signaling, resulting in the production of nitric oxide (NO). The ability of HDPs to inhibit NO production in LPS- or Lipid-A-activated RAW 264.7 macrophages was determined using different LPS structures. Both O111:B4 (smooth) and K12 (rough) LPS stimulated NO production at slightly differing concentrations. Less *E. coli* K12 LPS (5 ng/mL) was required to reach maximum NO production compared to LPS from *E. coli* O111:B4 (20 ng/mL). Large differences were observed between the neutralization activities of the tested HDPs. CATH-2 fully neutralized both rough and smooth LPS at a concentration of 5 µM (Figure 3A), while PMAP-36 and PMAP-23 required higher concentrations (10–20 µM) with small differences between K12 and O111:B4 LPS (Figure 3B,D). Interestingly, PR-39 did not have any effect on LPS-induced macrophage activation (Figure 3C).

Likewise, RAW 264.7 macrophages were stimulated with 50 ng/mL *E. coli* MPLA or DPLA. CATH-2 was most efficient in completely neutralizing DPLA (at 2.5 µM), followed by PMAP-36 (at 5 µM) and PMAP-23 (at 20 µM). In the case of MPLA, again, CATH-2 was most active, followed by PMAP-36 and PMAP-23 (Figure 4). Interestingly, for each HDP (except PR-39), large differences were observed between MPLA and DPLA in terms of neutralizing concentrations. CATH-2 and PMAP-36 neutralized DPLA with a 10-fold lower concentration than required for MPLA neutralization (Figure 4A,B), indicating involvement of the additional phosphate group of DPLA in interaction with the HDP. PR-39, as expected based on its lack of inhibition of LPS, did not have any effect on Lipid A-induced macrophage activation (Figure 4C).

### 2.4. Influence of LPS Structure on Binding Affinity of Host Defense Peptides

To investigate whether the observations made in LPS neutralization and the MBCs for the tested *E. coli* strains reflect the affinity between HDPs and LPS, binding was assessed using isothermal calorimetry (ITC). In case of O111:B4 and K-12, since the lipid A portion of the LPS is similar, any differences observed would be due to differences in the core and O-antigen. Different binding characteristics were observed for the different HDPs. CATH-2 binding to both LPS species was exothermic with relatively similar binding constants (Figure 5, Appendix A). PMAP-23 also showed roughly similar binding characteristics, although the dissociation constant was lower for K12 LPS. This would imply that the O-antigen of LPS plays a relatively small role in LPS binding to these HDPs. On the other hand, PMAP-36 exhibited a biphasic binding pattern to *E. coli* O111:B4 LPS with an initial exothermic but subsequent mainly endothermic binding. PR-39 also had mixed endothermic and exothermic binding for the same LPS. For both PMAP-36 and PR-39, no binding was observed for *E. coli* K12 LPS, indicating that the O-antigen is involved in the binding of these two HDPs and that this binding is more hydrophobic in nature compared to CATH-2 binding. Binding studies of the four HDPs with *E. coli* DPLA and MPLA were also performed, but these thermograms did not result in clear binding profiles for any of the peptide/lipid A combinations tested (Appendix A).

### 2.5. Polymyxin B Competition Assay

To obtain more insight into the LPS binding mechanism of tested HDPs, a competition LPS binding assay with dansyl-labelled polymyxin B (d-PMB) was performed as an additional method to determine the relative affinities of HDPs for LPS. This revealed that CATH-2 and PMAP-36 bind *E. coli* O111:B4 LPS tightly and that d-PMB could not displace these peptides. In contrast, d-PMB could compete with PR-39 and PMAP-23 in binding to the smooth LPS (Figure 6A). Similar trends were observed for LPS from *E. coli* K-12, but with smaller differences between peptides (Figure 6B).

### 2.6. Bacterial Membrane Permeabilization by Host Defense Peptides

To further characterize the antibacterial mechanism of these peptides, the kinetics of bacterial membrane permeabilization were investigated by flow cytometry. To distinguish between permeabilization of the inner and outer membranes, an *E. coli* strain expressing GFP in the cytoplasm and mCherry in the periplasm was studied [31].

Bacteria were incubated for 30 min with increasing concentrations of CATH-2, PMAP-36, PR-39, or PMAP-23 in the presence of Sytox in the medium. In this set-up, the release of mCherry indicates outer membrane permeabilization, the influx of Sytox indicates small perturbations of the inner membrane, and the release of GFP shows large inner membrane disruption. These experiments revealed that for both CATH-2 and PMAP-36, the Sytox influx was observed slightly before mCherry leakage. GFP leakage was simultaneous with mCherry leakage, indicating that the inner and outer membranes are disrupted almost simultaneously (Figure 7). Remarkably, at 10 µM and higher, the side scatter was observed to increase, which indicates morphological changes in the bacteria by CATH-2 and PMAP-36. For PR-39, no Sytox influx, mCherry, or GFP outflow was observed, confirming that this peptide does not affect the integrity of the bacterial membrane. PMAP-23 showed Sytox influx at higher concentrations, indicating small pores were formed when concentrations were sufficiently high.

In addition, membrane damage was assessed over time by exposing bacteria to 5 µM of CATH-2 or PMAP-36, or 20 µM of PR-39 or PMAP-23. This showed that within ten minutes, 80% of bacterial membranes were lysed by CATH-2 and PMAP-36. No membrane damage was observed over time by PR-39, confirming the results from the titration experiments. PMAP-23 showed a steady increase in Sytox influx, with 80% of bacterial membranes showing small pore formation after 45 min. Some leakage of mCherry and GFP was also observed after 45 min, indicating PMAP-23 is also capable of forming larger pores in both the inner and outer membrane. This affected bacterial morphology since the side scatter was also observed to increase over time by 20 µM of PMAP-23.

## 3. Discussion

In this study, *E. coli* was chosen as a model organism because it is one of the most common Gram-negative pathogens. Although many *E. coli* strains are present as commensals in, for example, the human gut, pathogenic strains are usually also present and can cause severe infections, leading to more than 2 million deaths each year [32]. Treatment of *E. coli* is based on antibiotic use, which has resulted in antibiotic resistance development against all common antibiotics. Most of these resistance mechanisms will not affect the sensitivity of HDPs because of their specific mechanisms. For example, ESBL-*E. coli* produces beta-lactamases that can hydrolyze penicillin [33], but these enzymes will not inactivate peptides. Due to the rise in multi-drug-resistant bacteria, colistin and polymyxin are used more frequently as last-resort antibiotics. This has resulted in the emergence of the first plasmid-mediated colistin resistance due to the MCR-1 gene [34]. Since MCR-1 affects LPS structure, this could have implications for the activity of HDPs as a potential alternative to antibiotics in the treatment of *E. coli* infections as well. Understanding how LPS mutations, not only MCR-1-mediated lipid A modifications but all structural aspects of *E. coli* LPS can affect the antibacterial and immunomodulatory activity of HDPs is essential for the further development of HDPs as novel anti-infective drugs.

The interactions of LPS with HDPs are well known and are important for two aspects of HDP activity. Firstly, many HDPs exert their antibacterial activity through membrane-active mechanisms and will first encounter LPS in their interaction with Gram-negative bacteria [35]. This interaction could either be an initial step in the antibacterial action of HDPs or it could retain HDPs in the outer membrane, limiting their ability to reach their potential targets: the inner membrane or intracellular molecules. Secondly, HDPs can function as LPS scavengers, preventing LPS from binding to Toll-like receptors on immune cells and preventing the potential toxic effects of LPS. To what extent the O-antigen or the phosphate groups of Lipid A of the LPS molecule are involved in binding HDPs and how that affects HDP dual activity was the novel aspect assessed in this study.

Interestingly, no obvious differences were observed in the antimicrobial potency of the four tested HDPs against *E. coli* O111:B4 containing smooth LPS and *E. coli* K12 containing rough LPS. In a similar type of study, despite the distinct membrane permeabilizing patterns, no differences in the bactericidal activity between rough and smooth *E. coli* LPS phenotypes were observed by two LL37-derived peptides, SAAP-148 and OP-145 [36]. This suggests the involvement of an LPS-independent antibacterial mechanism. In another study, the inner and outer core components of LPS were shown to have a major impact on *E. coli* susceptibility to some cationic antimicrobial peptides, indicating the importance of core sugars rather than the O-antigen in their case [19]. In the current study, no reduced susceptibility of MCR-1-positive *E. coli* was observed towards any of the HDPs, unlike colistin, which showed much lower activity against MCR-1-positive *E. coli*. This shows that HDP binding to these different LPSs is not fundamentally different or that binding LPS is not a part of the actual killing mechanism of HDPs. For PR-39, a cell-penetrating peptide that requires active uptake into the bacterial cytoplasm, this was already known, but the activity of CATH-2, PMAP-36, or PMAP-23 was not affected by the tested structural changes in the LPS molecule. In contrast, Colistin, whose antibacterial activity is based on its strong interaction with the phosphate moieties of Lipid A [37], was strongly affected by mutations in lipid A. This is in agreement with previous observations, in which no *mcr-1*-mediated cross-resistance towards some HDPs was observed [38,39], suggesting that the antibacterial mechanism of these peptides is different from colistin [40].

Our flow cytometry approach for detecting permeabilization of the inner and outer membranes shows that the HDPs quickly cross the outer membrane to penetrate and disrupt the inner membrane of *E. coli*. This is a very clear indication that, except for PR-39, the tested HDPs use LPS as an initial site for binding, and once the LPS barrier is traversed, the inner membrane can be disrupted immediately, and the rate of bacterial killing depends on the speed at which peptides can cross the LPS-containing outer membrane. A recent study using small cyclic hexapeptides shows exactly this concept using SPR and NMR. The authors describe that there is actually an optimum binding, or better said, an optimum dissociation, between the hexapeptides and LPS for the peptides to be able to reach and penetrate the inner membrane. Binding too strongly to LPS actually reduced the bacterial killing potency [41]. A similar theory can be applied to natural HDPs, and if the exact binding structure of HDPs and LPS is known, mutations toward stronger binding would lead to a lower MIC. What can be deduced from the current results is that especially the O-antigen, despite its anionic features, does not contribute significantly to the binding of the tested HDPs and thereby does not affect the protective value of the LPS layer.

When the ability of HDPs to neutralize LPS-induced macrophage activation was determined, interesting differences were observed. It is known that lipid A is the most immunogenic part of LPS, and rationally, the immune system has evolved to specifically recognize this conserved part of LPS [42]. When LPS is presented to TLR4 through intermediate transfer proteins LBP, CD14, and MD2, the downstream signaling cascade results in the expression of multiple pro-inflammatory cyto- and chemokines [43]. CATH-2 and PMAP-36 strongly neutralized both O111:B4 and K12 LPS-induced activation of macrophages, while PMAP-23 showed intermediate neutralization. The comparable activity for both LPS structures is in line with the fact that the lipid A moiety is similar for both O111:B4 and K12 LPS. Differences were observed for the neutralization of DPLA and MPLA. Having only one phosphate group on the lipid A moiety apparently reduced the capacity of HDPs to neutralize subsequent macrophage activation, indicating that the phosphate groups are involved in binding these HDPs [44,45]. The observed effect of the lipid A phosphate group in our experiments suggests direct LPS-peptide interaction as a major neutralization mechanism, at least for the tested HDPs. Additionally, HDPs could theoretically also interfere with LBP, MD-2, or CD14, the adaptor proteins required for presenting LPS to TLR4. Although no indications for this phenomenon have been described yet, some molecular simulation experiments provide a basis for this possibility [46]. However, the possibility of cell surface or intracellular targets for these HDPs cannot be ruled out completely, as only a co-incubation setup was tested in this study.

LPS binding was assessed using ITC for CATH-2, PMAP-36, PR-39, and PMAP-23, which actually showed clear differences in binding mode between these different HDPs. CATH-2 and PMAP-23 showed exothermic binding, while PMAP-36 and PR-39 showed mixed exothermic-endothermic binding (Figure 5) for *E. coli* O111:B4 (smooth) LPS. This exothermic binding of CATH-2 with similar LPS species has been shown before, is via hydrogen bonding, and can be hydrophobic at higher concentrations of peptide [24]. PMAP-36 and PR-39 showed a biphasic binding pattern where exothermic and endothermic binding simultaneously or sequentially took place. A similar biphasic binding pattern for PMAP-36 has been shown previously, which could indicate initial binding depending on hydrogen bonding followed by more hydrophobic interaction when the peptide starts to accumulate [24]. This binding mode of PMAP-36 with *E. coli* O111:B4 LPS differs from that of the other tested HDPs, which are more ionic in nature. Interestingly, no binding was seen for PMAP-36 and PR-39 with *E. coli* K12 rough LPS, which is overall more hydrophobic than smooth LPS, which suggests that the interactions of these two HDPs with LPS are mainly with the O-antigen part of LPS. Most importantly, though, no correlation could be found between the LPS binding of HDPs and the LPS neutralizing or antibacterial activity of the same HDPs. This lack of correlation might be explained by the differences in assay conditions. For ITC, relatively high concentrations of LPS, Lipid A, and HDP are required in order to produce enough heat to be detected. These higher concentrations mean that LPS will be present in micelles instead of the soluble monomeric form in the macrophage activation assays. The binding of HDP to micellar LPS can be very different from that of soluble LPS. Also, components of cell culture medium, such as LBP or possibly soluble CD14, could affect the state of LPS presented to HDPs. Furthermore, binding LPS does not necessarily lead to neutralization, as shown previously for the synthetic peptide Murepavadin [47]. This could, for example, explain why PR-39 can bind LPS in the ITC experiments (and in the replacement-PMB competition assay) but is not able to neutralize LPS.

For Lipid A, the phase diagram is even more complex. Lipid A can form lamellar structures in water and has five different three-dimensional aggregation states, including micellar (as LPS), bilayer-based phases, such as the hexagonal and inverted hexagonal phases, and additionally a non-lamellar cubic phase [48]. The transition between phases is dependent on many variables, such as pH and temperature, but also, for example, the nature of the counter ion (mono or divalent). It is beyond the scope of the current study to determine in what phase MPLA and DPLA were actually present, but it is very likely that the aggregated structure of Lipid A is important for its interaction with HDPs. Finally, another disadvantage of ITC is that only heat production (or heat input) upon binding is measured, and other binding characteristics are calculated from the measured heat data. If a binding reaction is mainly driven by entropy instead of enthalpy (heat production), this will not be detected in ITC measurements. For example, no clear binding was seen for PMAP-36 binding to *E. coli* K12 LPS based on heat production, but that does not completely rule out that entropy-based binding is still occurring.

For two other LPS-binding peptides, Pardaxin and hLF11, the interaction with LPS was resolved by NMR [49,50]. Both peptides interacted with the lipid A part of LPS (in micelles), where three positively charged residues of the peptide aligned with the phosphorylated glucosamines of lipid A. Additionally, hydrophobic amino acid residues were in close contact with the acyl chains of lipid A. For CATH-2, a similar binding motif was predicted based on a similar basic/hydrophobic residue pattern in the peptide [51]. This prediction corresponds well with the obtained ITC data, which show similar binding to rough and smooth LPS for CATH-2. Interestingly, PMAP-23 also exhibited similar binding to rough and smooth *E. coli* LPS yet did not bind (unlike CATH-2) in earlier studies to rough LPS from *Salmonella minnesota* R595 [28]. Lipid A of *E. coli* and *Salmonella* Minnesota LPS are supposedly identical with respect to phosphorylated glucosamines and the presence of six acyl chains [52], which indicates that this might not be the major binding site for PMAP-23.

Altogether, these results suggest that LPS binding is not integral to the antibacterial mechanism of the HDPs tested in this study and that LPS modifications hardly affect the antibacterial activity. However, subsequent neutralization of LPS-induced inflammation can be affected by LPS modifications.

## 4. Materials and Methods

### 4.1. Peptide Synthesis

Peptides CATH-2, PMAP-36, PR-39, and PMAP-23 were synthesized by Fmoc solid-phase synthesis at China Peptides (CPC Scientific, Sunnyvale, CA, USA) and at ACTA (Amsterdam, The Netherlands). All peptides were purified to a purity of >95% by reverse-phase high-performance liquid chromatography. The sequences and characteristics of the peptides are shown in Table 1.

### 4.2. Bacterial Strains

*E. coli* K-12 (ATCC 10798), *E. coli* O111:B4 (clinical isolate, University Medical Center Groningen, Groningen, The Netherlands), *E. coli* NCTC 13846 (MCR-1 positive), and *E. coli* 078 (clinical isolate, University Medical Center Utrecht, Utrecht, The Netherlands) were used throughout this study. All *E. coli* strains were grown on tryptic soy agar (TSA) plates (Oxoid Ltd., Basingstoke, Hampshire, UK). Liquid cultures were grown in lysogeny broth (LB) containing 1% yeast extract (Becton, Dickinson and Company, Sparks, NV, USA), 1% NaCl (Merck, Darmstadt, Germany), and 0.5% tryptone (Becton, Dickinson and Company, Franklin Lakes, NJ, USA) or Mueller-Hinton Broth (MHB, Becton, Dickinson and Company).

### 4.3. Cell Culturing

RAW 264.7 cells (ATCC TIB-71) were cultured in Dulbecco’s Modified Eagle Medium (DMEM) (Gibco, Thermo Fisher Scientific, Waltham, MA, USA) supplemented with 10% fetal calf serum (FCS) (Bodinco B.V., Alkmaar, The Netherlands) and 100 units/mL Penicillin and 100 µg/mL Streptomycin at 37 °C, 5.0% CO_2_. For LPS/Lipid A stimulation studies, 5 × 10^4^ cells/well were first seeded in a 96-well plate and kept for 24 h to adhere.

### 4.4. Track Dilution Assay

Bacterial killing by HDPs was assessed using track dilution assays, as described before [55]. In short, 10^6^ colony-forming units (CFU)/mL of bacteria were incubated with different concentrations of peptides for 3 h at 37 °C in a U-bottom microtiter plate (Corning, New York, NY, USA). After incubation, 10-fold dilutions were prepared using the corresponding medium, and 10 µL of each dilution was plated on appropriate agar plates. Plates were incubated at 37 °C, and colonies were counted after 24 h. Minimal bactericidal concentration (MBC) was defined as ≤500 CFU/mL.

### 4.5. LPS/Lipid A Neutralization Assays

RAW 246.7 cells were seeded (5 × 10^4^ cells/well) in a 96-well plate and left at 37 °C for overnight adherence. Then, cells were stimulated with 20 ng/mL LPS originating from *E. coli* O111:B4 (InvivoGen, San Diego, CA, USA), or 5 ng/mL LPS originating from *E. coli* K-12 (InvivoGen), and also with 50 ng/mL Lipid A, MPLA, or DPLA from *E. coli* F583 (Sigma, St. Louis, MO, USA) with or without 0–20 µM HDPs in DMEM for 24 h. After incubation, the nitrite production in the supernatant was measured using the Griess assay [56]. Briefly, 50 µL cell culture supernatant was mixed with 50 µL 1% Sulfanilamide (Sigma-Aldrich, Zwijndrecht, The Netherlands) and incubated at room temperature in the dark for 5 min. Then, 50 µL of 0.1% N-(1-Naphthyl)ethylenediamine dihydrochloride (VWR International B.V., Amsterdam, The Netherlands) was added and incubated at room temperature in the dark for 5 min. Sodium nitrite (Sigma) was used as a standard to accurately determine the nitrite concentration in the cell supernatant. Samples were measured at 590 nm using a FLUOstar Omega microplate reader (BMG Labtech GmbH, Ortenberg, Germany).

### 4.6. Isothermal Titration Calorimetry (ITC)

ITC was performed with a Low Volume NanoITC (TA Instruments-Waters LLC, New Castle, DE, USA). The 50 µL syringe was filled with 200 µM peptide in 1:3 H_2_O:phosphate buffered saline (PBS) for titration into 164 µL of 62.5 µM LPS or 25 µM Lipid A in 1:3 H_2_O:PBS, unless stated otherwise. Titrations were incremental, with 2 µL injections at 300 s intervals. Experiments were performed at 37 °C. The data were analyzed with the NanoAnalyze software (TA Instruments-Waters LLC).

### 4.7. Dansyl-Polymyxin B Competition Assay

Different concentrations of peptide (25 µL) were incubated with 15 µg/mL of LPS (25 µL) in a flat-bottom 96-well plate at 37 °C for 30 min. Afterwards, 50 µL of 8 µM dansyl-labelled polymyxin B was added (end concentration of 4 µM), mixed, and fluorescence was determined immediately using the Fluostar Omega. Samples were excited at 340 nm, and the signal was measured at 490 ± 10 nm. Signals were corrected for dansyl-polymxyin B background. Bound d-PMB gives a high fluorescent signal at 485 nm, which decreased with increasing peptide concentrations, indicating less d-PMB was able to bind. This was converted to percentages of bound d-PMB.

### 4.8. Flow Cytometry

Recombinant *E. coli* expressing mCherry in the periplasm and Green Fluorescent Protein (GFP) in the cytoplasm (_Peri_mCherry/_cyto_GFP) was prepared as previously described [31]. Bacteria were grown overnight in LB medium containing 100 µg/mL ampicillin. The next day, subcultures were grown to mid-log phase (optical density, OD_600_ of approx. 0.5), washed, and resuspended to an OD_600_ of approx. 1 in RPMI supplemented with 0.05% human serum albumin (RPMI-HSA). All further incubations were carried out in RPMI-HSA. Bacterial cultures with an OD_600_ of approx. 0.05 were mixed with 1 µM Sytox Blue Dead Cell Stain (Thermo Fisher Scientific, Waltham, MA, USA) and exposed to a concentration range of HDPs for 30 min at 37 °C. For the kinetic experiments, bacteria were mixed with the HDPs (concentrations indicated in figure legends) for variable times, up to 45 min, and incubated at 37 °C. After the incubations, bacteria were diluted ten times, after which the Sytox blue, mCherry, and GFP intensities were analyzed by flow cytometry (MACSQuant). Data were analyzed in FlowJo, where the percentage of mCherry and GFP-negative or Sytox positive bacteria was determined by gating on the buffer control.

### 4.9. Statistical Analysis

Statistical analysis was performed using GraphPad Prism version 9.3.1 (471). All quantitative measurements were conducted as three independent measurements, which were made in duplicate. One-way ANOVA was used as an appropriate method to test for significant differences, followed by Sidak’s multiple comparison test. A *p* value of <0.05 was considered significant.

## 5. Conclusions

Gram-negative outer membrane remodeling in terms of LPS structural modification is critical for the interaction with cationic HDPs. However, owing to different LPS binding and membrane permeabilizing mechanisms of HDPs, the effect of these modifications on bacterial killing and membrane-free LPS interaction could vary. The structural features of *E. coli* LPS examined in this study only seem to have a very limited effect on the antibacterial activity of the tested HDPs. The O-antigen of *E. coli* LPS, although in many ways a very important virulence factor, was not majorly affecting HDPs antibacterial activity. Also, the number of free phosphate groups of lipid A, the main binding site for HDPs, did not have a considerable impact on the susceptibility of *E. coli* to the HDPs, although actual binding to some HDPs seemed to be affected. The anti-endotoxin activity of HDPs, on the other hand, depended on the number of phosphate groups in LPS. Therefore, these LPS modifications that do not affect HDP antibacterial activity could still impact the effectiveness of HDPs in fighting infection in vivo because the dampening of the potential damaging pro-inflammatory response towards LPS is reduced. It can be concluded from these results that LPS binding could play a role in the HDP anchorage in the Gram-negative outer membrane and could be affected by LPS structural modification, but it does not solely contribute to the killing mechanism. Furthermore, the LPS-neutralizing interaction of these HDPs does not necessarily reflect the lethal action on the bacterial outer membrane.

## 6. Future Trends

To further understand the role of LPS in either hindering or enabling the activity of HDPs, focus should also include other types of LPS modifications, such as acylation patterns. The in-depth biophysical characterization of these HDPs in the presence of bacterial and model membranes containing LPS structural mutants could provide better insight about the mode of binding and the effects on both activities of HDPs. In exploring the potential of HDPs as alternatives to conventional antibiotics, the anti-endotoxin activity and how it is affected by LPS structure should be a substantial part of future research.

## Figures and Tables

**Figure 1 pharmaceuticals-16-01485-f001:**
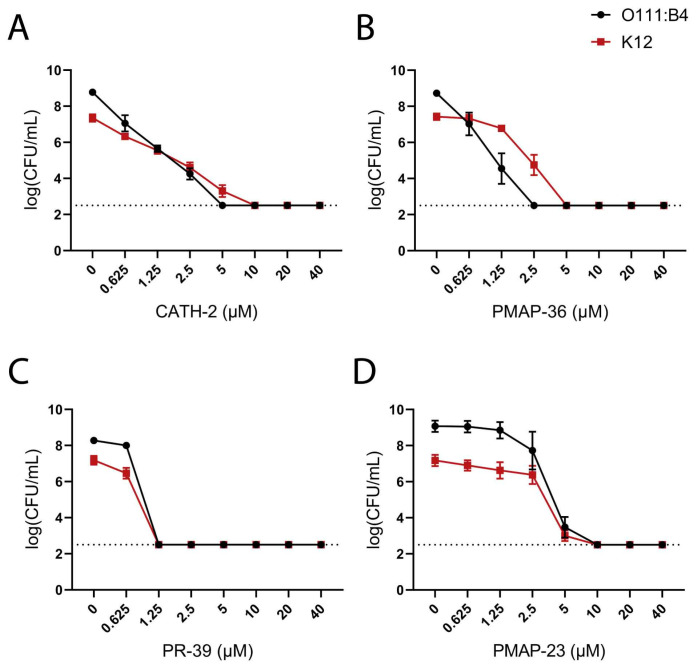
Determination of MBC values of HDPs against *E. coli* strains (O111:B4 or K-12). MBC values of CATH-2 (**A**), PMAP-36 (**B**), PR-39 (**C**), and PMAP-23 (**D**) were determined by colony count assay. Surviving bacterial colonies were detected after incubation with HDPs for 3 h in MHB. Shown are the mean ± SEM of three independent experiments; the dashed line shows the detection limit of the assay.

**Figure 2 pharmaceuticals-16-01485-f002:**
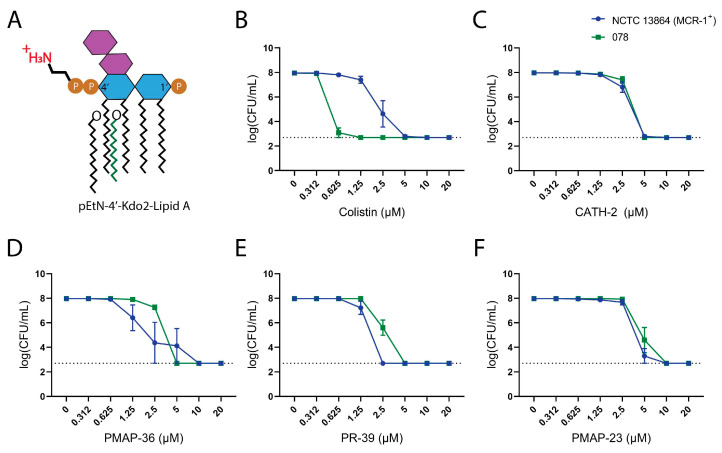
Determination of MBC values of HDPs against MCR-1^+^ *E. coli* (**A**) MCR-1^+^ bacteria have a Phosphoethanolamine (pEtN) group attached to 1′or 4′-Phosphate of Lipid A in the LPS. MBC values of Colistin (**B**), CATH-2 (**C**), PMAP-36 (**D**), PR-39 (**E**), and PMAP-23 (**F**) for *E. coli* strains (NCTC 13864 or 078) were determined by colony count assay. Surviving bacterial colonies were detected after incubation with HDPs for 3 h in MHB. Shown are the mean ± SEM of three independent experiments; the dashed line shows the detection limit of the assay.

**Figure 3 pharmaceuticals-16-01485-f003:**
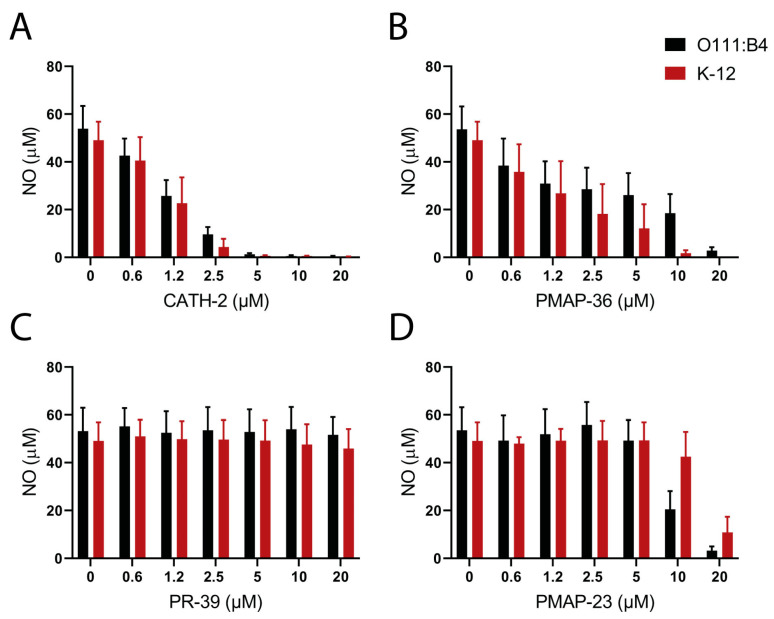
Neutralization of LPS-induced macrophage activation by HDPs. RAW264.7 cells were stimulated with *E. coli* LPS (O111:B4 or K-12) in the presence of different concentrations of CATH-2 (**A**), PMAP-36 (**B**), PR-39 (**C**), or PMAP-23 (**D**). NO production was measured by the Griess assay in duplicates. Shown are mean + SEM of three independent experiments.

**Figure 4 pharmaceuticals-16-01485-f004:**
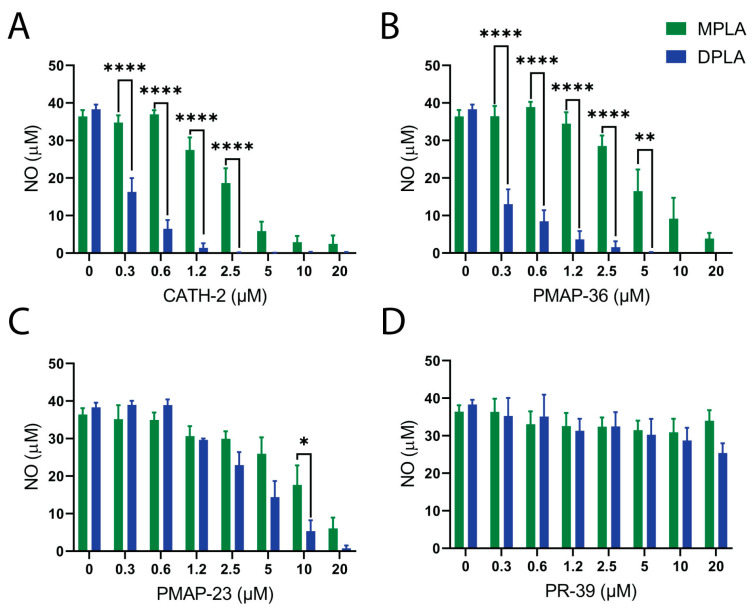
Neutralization of Lipid A-induced macrophage activation by HDPs. RAW264.7 cells were stimulated with *E. coli* F583 Lipid A (MPLA or DPLA) in the presence of different concentrations of CATH-2 (**A**), PMAP-36 (**B**), PR-39 (**C**), or PMAP-23 (**D**). NO production was measured by the Griess assay in duplicates. Shown are mean + SEM of three independent experiments. * *p* < 0.05; ** *p* < 0.01; **** *p* < 0.0001.

**Figure 5 pharmaceuticals-16-01485-f005:**
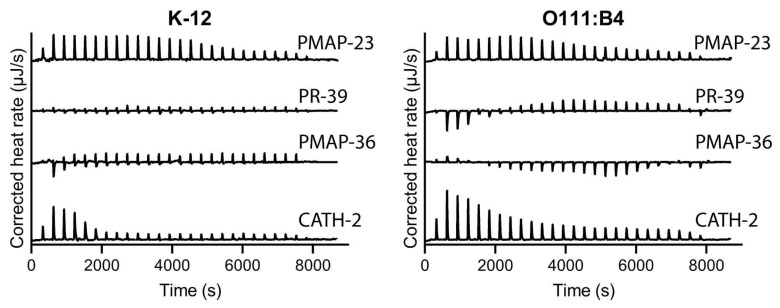
ITC spectra of the interaction between LPS species and HDPs. Approximately 200 µM of CATH-2, PMAP-36, PR-39, or PMAP-23 was titrated into an *E. coli* LPS (O111:B4 or K-12) solution (0.5 mg/mL, 0.25 mg/mL for *E. coli* O111:B4 and CATH-2) and heat rates were recorded. Shown is a representative of two measurements.

**Figure 6 pharmaceuticals-16-01485-f006:**
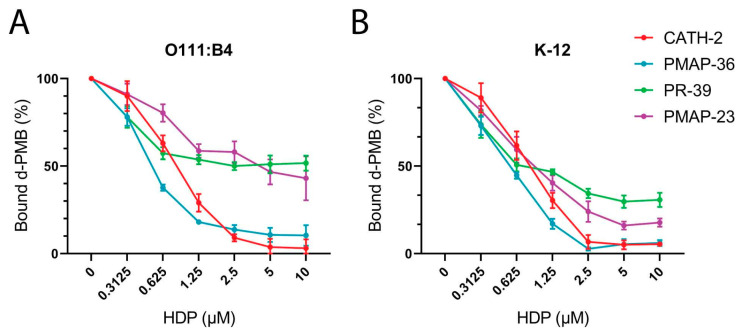
LPS binding competition assay with dansyl-labelled polymyxin B (d-PMB) O111:B4 (**A**) and K12 (**B**) LPS were incubated with increasing concentrations of HDPs. D-PMB (8 µM) was added afterwards, and the amount of bound d-PMB (shown as a percentage) was measured with the fluorescent signal at 490 nm. Shown are the means ± SEM of three independent experiments.

**Figure 7 pharmaceuticals-16-01485-f007:**
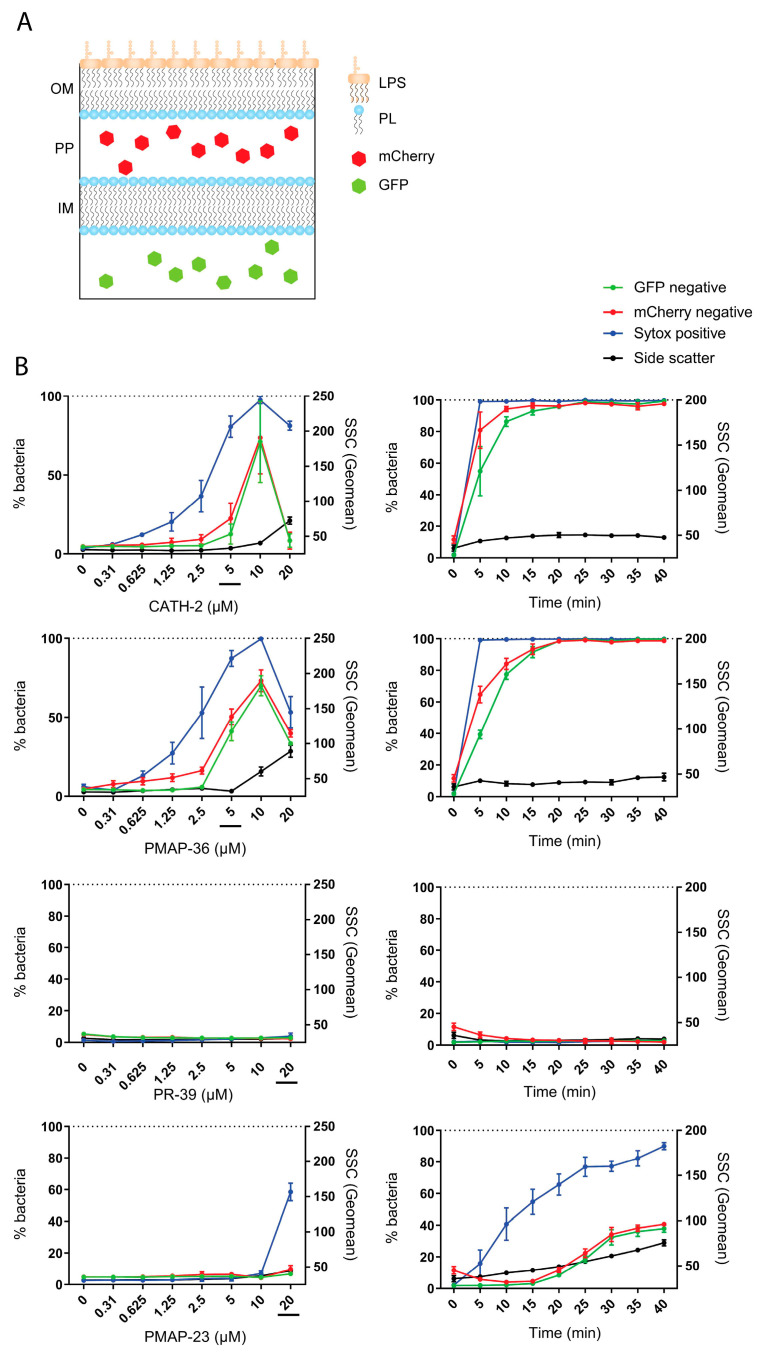
Flow cytometry and membrane permeabilization assays were used to determine the antibacterial mechanisms of CATH-2, PMAP-36, PR-39, and PMAP-23. (**A**) An *E. coli* strain expressing mCherry in the periplasm (PP) and Green Fluorescent Protein (GFP) in the cytoplasm was used to assess outer membrane (OM) and inner membrane (IM) damage. LPS, lipopolysaccharide; PL, phospholipid. (**B**) Increasing concentrations of HDPs (**left**) were studied, and one concentration of each HDP (underlined in the concentration graph) was also assessed over time (**right**). Sytox influx was also measured to observe membrane destabilization and side scatter to study bacterial morphology upon treatment with HDPs. Shown are the means ± SEM of three independent experiments.

**Table 1 pharmaceuticals-16-01485-t001:** Sequence, number of amino acids (No. aa), and charge and mass of studied peptides [24,27,28,53,54].

Peptide	Sequence	No. aa	Charge	Mass (Da)
CATH-2	RFGRFLRKIRRFRPKVTITIQGSARF-NH_2_	26	9+	3208
PMAP-36	Ac-GRFRRLRKKTRKRLKKIGKVLKWIPPIVGSIPLGCG	36	13+	4198
PR-39	RRRPRPPYLPRPRPPPFFPPRLPPRIPPGFPPRFPPRFP	39	10+	4721
PMAP-23	RIIDLLWRVRRPQKPKFVTVWVR	23	6+	2963

## Data Availability

The data presented in this study are available on request from the corresponding author. The authors can confirm that all relevant data are included in this article.

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
