# Peer review of "Effects of Escherichia coli LPS Structure on Antibacterial and Anti-Endotoxin Activities of Host Defense Peptides"

_pharmaceuticals, 2023, doi:10.3390/ph16101485_

Round 1

Reviewer 1 Report

  - The abstract needs to be rewritten entirely. The abstract is very generic and cliche and does not provide any significant novelty or promising results. Please include important results and clearly state what the significance and novelty of the manuscript are.

 - Introduction: I suggest improving the literature review to provide the background presentation.

- The novelty of this work should add to abstract and text.

- The results are presented like a progress report without including in-depth discussion. I recommend a more profound discussion about the results instead of a simple presentation of the outcomes of the experiment.

- The conclusion part is too weak. Add more in-depth discussion".

- Did you take any protein analyses such as western blots and...?

- Did the authors consider some specific points for Influence of LPS structure on binding affinity of host defense peptides?

Author Response

Note: The line numbers in our answers refer to the new manuscript without tracked changes.

Reviewer 1

1) The abstract needs to be rewritten entirely. The abstract is very generic and cliche and does not provide any significant novelty or promising results. Please include important results and clearly state what the significance and novelty of the manuscript are.

Response: We agree that the abstract was very generic and did not really address what is novel and interesting about this study. We have now completely rewritten the abstract, removing cliches and highlighting the novelty of the study, the combined effect of LPS modifications on both HDPs  immunomodulatory and antibacterial activity. Line 17-37.

 2) Introduction: I suggest improving the literature review to provide the background presentation

Response: We extended the introduction with substantial new background information and references. Page 2, Line 54-58;  78-83, 97-100, and more…

3) The novelty of this work should add to abstract and text .

Response: We added the novelty, the focus on the dual activity of HDPs and how that is affected by LPS modifications. to the abstract Line 22-24, and the main text.  Line 284 -293.

4) The results are presented like a progress report without including in-depth discussion. I recommend a more profound discussion about the results instead of a simple presentation of the outcomes of the experiment.

Response: Major parts of the discussion were re-written and updated with more in-depth discussion of results, supported by references. Among others: page 10, Line 296-302, 312-313; , 319-328..

5) The conclusion part is too weak. Add more in-depth discussion"

Response: The conclusion section has been re-written. It has been updated with more generalized statements with more clear discussion points based on our outcomes. Page 14, Line 492-495; 502-509.

6) Did you take any protein analyses such as western blots and...?

Response: We only look at antibacterial activity and immunomodulation. Since we know that bacterial killing of these peptides is very fast (within minutes, we don’t expect to see major changes in the protein composition of the bacteria.

7) Did the authors consider some specific points for Influence of LPS structure on binding affinity of host defense peptides?

Response: We focused on the regular read out parameters that isothermal titration calorimetry provides: Kd, enthalpy, and entropy as shown in suppl file S3 and discussed in the text. Proper interpretation of exact binding modes of HDP and would require NMR studies which would be very interesting as a follow up story. We discussed a bit of the binding modes in the discussion parts in paragraph from Line 349-390.

Reviewer 2 Report

Dear Author, I reviewed the manuscript (pharmaceuticals-2640787) entitled Effects of E. coli LPS structure on antibacterial and anti-endotoxin activities of host defense peptides. This manuscript presents relevant information about LPS's role in endotoxin activities and host defense peptides. However, some sections of the presented data can be improved. For this reason, I consider that this manuscript needs minor changes to be considered for publication in this journal. 

Additional comments.

Highlight the advantages of evaluating the LPS effect on bacterial killing.

Check paragraphs extension in this manuscript. 

Include an experimental design that contains statistical factors and variables of response in the statistical analyses applied to the findings of this research.

Compare the obtained findings with similar assays where other peptides or similar antibacterial structures interact with E. coli LPS. 

Include future trends to keep working with the obtained data. 

Try to conclude with a general statement of the most relevant part of this study.

Additional specific comments.

The manuscript must include and highlight how specific characteristics of E. coli LPS influence the effectiveness of host defense peptides in their antibacterial and anti-endotoxin functions.

The state of the art is highly relevant in the microbiology field. However, the novelty of this research must detail how the bacterial cell structure and the host's defense mechanisms potentially shed light on new strategies for combatting E. coli infections.

 The findings of this research need in-depth discussion on the practical implications for developing treatments or interventions against E. coli infections. LPS structural characteristics must be discussed.

The experimental design needs to include statistical factors and variables of response in the statistical analyses applied to the findings of this research.

Strengthening the conclusion section with a more profound discussion is imperative to provide a satisfying conclusion to the research.

These improvements will significantly enhance the manuscript's overall quality and contribution to the field.

Author Response

Note: The line numbers in our answers refer to the new manuscript without tracked changes.

Dear Author, I reviewed the manuscript (pharmaceuticals-2640787) entitled Effects of E. coli LPS structure on antibacterial and anti-endotoxin activities of host defense peptides. This manuscript presents relevant information about LPS's role in endotoxin activities and host defense peptides. However, some sections of the presented data can be improved. For this reason, I consider that this manuscript needs minor changes to be considered for publication in this journal. 

Additional comments.

1) Highlight the advantages of evaluating the LPS effect on bacterial killing.

Response: The introduction and discussion sections have been updated with importance of evaluating effect of LPS structure on bacterial killing.  Lines 66-70; 78-82; 97-98.97; 268-283

2) Check paragraphs extension in this manuscript.

Response: It is unclear to us what to check about the paragraphs exactly, but we checked whether the text would benefit from removal or extensions of certain paragraphs. And have removed parts where the text was too extended for the message of those specific paragraphs. For example we removed a few unnecessary lines on the anti-endotoxin activity of the human  LL-37 cause that didn’t add anything to the discussion of our current data.

3) Include an experimental design that contains statistical factors and variables of response in the statistical analyses applied to the findings of this research.

Response: The materials and methods sections contains the description of the statistical analyses used. And each figure legend explains the number of experiments and what is shown in the figure, including significance levels.

4) Compare the obtained findings with similar assays where other peptides or similar antibacterial structures interact with E. coli LPS

Response: Discussion has been updated with more comparisons of obtained results with similar type of studies. Line 296-302, 312-313; , 319-328

5) Include future trends to keep working with the obtained data.

Response: Future trends have been included as a separate section after conclusions. Lines 511-518.

6) Try to conclude with a general statement of the most relevant part of this study.

Response: The conclusion part has been updated with more generalized statements with more clear discussion based on our outcomes. Line 492-495; 502-509.

Additional specific comments.

7) The manuscript must include and highlight how specific characteristics of E. coli LPS influence the effectiveness of host defense peptides in their antibacterial and anti-endotoxin functions .

Response: The introduction and discussion sections have been updated with the significance of studying E. coli LPS structure in regard to the host defense peptides activities. Lines 73, 80, parts of 230-240;  parts of 268-283

8) The state of the art is highly relevant in the microbiology field. However, the novelty of this research must detail how the bacterial cell structure and the host's defense mechanisms potentially shed light on new strategies for combatting E. coli infections ‘

Response: Principally we used E. coli only as a model organism for Gram negative bacteria in general. Mainly because it is a well-studied bacterium but also because of the importance of E.coli infections in general and the difficulty of treating multidrug-resistant E.coli at the moment. In the discussion we now added a paragraph on E. coli infections and treatment. Line 268-283. The importance of the novelty is especially highlighted in the last 2 sentences of this paragraph.

9) The findings of this research need in-depth discussion on the practical implications for developing treatments or interventions against E. coli infections. LPS structural characteristics must be discussed.

Response: Major parts of the discussion were re-written and updated with more in-depth discussion of results, supported by references. More points to discuss the importance of studying the effect of LPS structure on host defense peptides’ activities against E. coli infection have been added. Page 9, 268-283; parts of 294-313

10) The experimental design needs to include statistical factors and variables of response in the statistical analyses applied to the findings of this research.

Response: See above point 3 ( I presume this reviewers remark was accidently copied ).

11) Strengthening the conclusion section with a more profound discussion is imperative to provide a satisfying conclusion to the research.

Response: Conclusions section was updated with more clear statements based on the key discussion points about our outcomes.  Line 492-502.

These improvements will significantly enhance the manuscript's overall quality and contribution to the field.

Response: we agree, thank you for the valuable remarks.

Reviewer 3 Report

Authors investigated mechanism of activity of four peptides with known antibacterial activity. Hypothesis was focused on an interaction between investigated peptides and lipopolysaccharides (LPS) as a source of antibacterial activity and generally is negatively verified. The work is still interesting, but I have some minor comments.

Table 1 – A short characteristic of used peptides should be expanded with molecular mass and  zetapotential data for each peptide. I suggest to start data presentation with this table to improve text clarity and to help compare results between peptides.

At page 8 line 230 authors write: “ After a short delay also GFP leakage was observed, indicating small pores were formed first, followed by larger disruptions in the outer membrane and finally the inner membrane (Figure 7).” For both CATH-2 and PMAP-36, GFP and mCherry leakage is simultaneous. It indicates a parallel breach of both membranes.

Authors use molar concentrations of peptides and antibiotics, it is fine, but I do suggest to add Fig 1 and Fig 2 versions with microg/mL scale to supporting information, to help comparison with other available data.

Author Response

Note: The line numbers in our answers refer to the new manuscript without tracked changes.

Reviewer 3

Open Review

Authors investigated mechanism of activity of four peptides with known antibacterial activity. Hypothesis was focused on an interaction between investigated peptides and lipopolysaccharides (LPS) as a source of antibacterial activity and generally is negatively verified. The work is still interesting, but I have some minor comments.

1) Table 1 – A short characteristic of used peptides should be expanded with molecular mass and  zetapotential data for each peptide. I suggest to start data presentation with this table to improve text clarity and to help compare results between peptides.

Response: We added peptide masses to table 1, as requested, but were unclear about the zeta potential. This is, to our knowledge not a characteristic of a specific peptide, but usually the effect of peptides on, for example a nanoparticle or bacterium is determined. This is also affected by concentration, buffer conditions , pH etc., so cannot be captured in a single fnumber in a table.

2) At page 8 line 230 authors write: “ After a short delay also GFP leakage was observed, indicating small pores were formed first, followed by larger disruptions in the outer membrane and finally the inner membrane (Figure 7).” For both CATH-2 and PMAP-36, GFP and mCherry leakage is simultaneous. It indicates a parallel breach of both membranes

Response: The reviewer is absolutely correct, we corrected this statement in the text Line 249-252.

3) Authors use molar concentrations of peptides and antibiotics, it is fine, but I do suggest to add Fig 1 and Fig 2 versions with microg/mL scale to supporting information, to help comparison with other available data.

Response: We added these adapted figures and refer to them in the text as supplementary figures as requested,

Round 2

Reviewer 1 Report

All comments are fully addressed and manuscript accept in present form.